# Exploiting the Dixon Method for a Robust Breast and Fibro-Glandular Tissue Segmentation in Breast MRI

**DOI:** 10.3390/diagnostics12071690

**Published:** 2022-07-11

**Authors:** Riccardo Samperna, Nikita Moriakov, Nico Karssemeijer, Jonas Teuwen, Ritse M. Mann

**Affiliations:** 1Department of Medical Imaging, Radboudumc, 6525 GA Nijmegen, The Netherlands; n.moriakov@nki.nl (N.M.); nico.karssemeijer@gmail.com (N.K.); j.teuwen@nki.nl (J.T.); ritse.mann@radboudumc.nl (R.M.M.); 2Department of Radiology, The Netherlands Cancer Institute (NKI), 1066 CX Amsterdam, The Netherlands; 3Department of Radiation Oncology, The Netherlands Cancer Institute (NKI), 1066 CX Amsterdam, The Netherlands; 4ScreenPoint Medical BV, 6525 EC Nijmegen, The Netherlands

**Keywords:** breast, segmentation, deep learning, MRI, data-centric AI

## Abstract

Automatic breast and fibro-glandular tissue (FGT) segmentation in breast MRI allows for the efficient and accurate calculation of breast density. The U-Net architecture, either 2D or 3D, has already been shown to be effective at addressing the segmentation problem in breast MRI. However, the lack of publicly available datasets for this task has forced several authors to rely on internal datasets composed of either acquisitions without fat suppression (WOFS) or with fat suppression (FS), limiting the generalization of the approach. To solve this problem, we propose a data-centric approach, efficiently using the data available. By collecting a dataset of T1-weighted breast MRI acquisitions acquired with the use of the Dixon method, we train a network on both T1 WOFS and FS acquisitions while utilizing the same ground truth segmentation. Using the “plug-and-play” framework nnUNet, we achieve, on our internal test set, a Dice Similarity Coefficient (DSC) of 0.96 and 0.91 for WOFS breast and FGT segmentation and 0.95 and 0.86 for FS breast and FGT segmentation, respectively. On an external, publicly available dataset, a panel of breast radiologists rated the quality of our automatic segmentation with an average of 3.73 on a four-point scale, with an average percentage agreement of 67.5%.

## 1. Introduction

Breast dynamic contrast-enhanced magnetic resonance imaging (DCE-MRI) has higher sensitivity for breast cancer than digital mammography, particularly for screening, as multiple studies in the literature have shown [1]. The higher sensitivity of DCE-MRI has led to the widespread adoption of this imaging tool in the screening of women at very high risk of developing breast cancer. Breast DCE-MRI is also gaining more attention as an additional screening tool for women at average or only slightly increased risk due to the mounting evidence of its effectiveness. One example of women at average risk of developing breast cancer includes women with extremely dense breasts. The DENSE trial has recently proven the effectiveness of using breast DCE-MRI as an additional imaging modality to reduce the incidence of interval cancer in women with extremely dense breasts [2]. Recently, the European Society of Breast Imaging (EUSOBI) has updated its recommendations for women with extremely dense breasts by indicating dynamic DCE-MRI as the best tool for screening in this population [3].

Breast density (BD), defined as the ratio between fat and fibro-glandular tissue (FGT) in the breast, is a recognized risk factor for the future development of breast cancer. Breast MRI allows the more accurate quantification of breast density compared to digital mammography, thanks to its higher dimensionality and higher contrast between fat and FGT. A robust breast density quantification requires an accurate segmentation of the breast and the FGT areas. The correct delineation of the boundary between the pectoral muscle and the breast tissue, especially in dense breast cases, is the main challenge affecting the accuracy of breast segmentation, while intensity inhomogeneities make the FGT segmentation non-trivial. In clinical practice, radiologists still rely on the visual assessment of breast density using the Breast Imaging Reporting and Data System (BI-RADS) [4], which has been shown to suffer from strong inter- and intra-reader variability [5]. While automatic software for breast density estimation is already available for mammography, only limited attempts have been made at creating a robust breast density calculation tool for breast MRI.

Automatic breast and FGT segmentation in breast MRI has been addressed by several authors in the literature, using methods ranging from traditional computer vision approaches to deep learning methods. Deep learning methods have shown great results at automating different medical imaging tasks, consistently outperforming traditional methods. For medical imaging segmentation, in particular, the U-Net architecture [6] has been the de facto standard. Dalmis et al. [7] used a 2D U-Net to segment breast and FGT in breast DCE-MRI acquisitions without fat suppression (WOFS), showing better results compared to an atlas-based approach [8] and a sheetness-based approach [9]. Zhang et al. [10] proved the robustness of the U-Net architecture over a test set composed of WOFS breast DCE-MRI acquired from different scanners. More recently, Huo et al. [11] made use of the popular open-source framework nnUNet [12] to confirm the great potential of deep learning approaches for the task of breast and FGT segmentation in breast DCE-MRI acquisitions with fat suppression (FS).

The literature reviewed so far mainly relied on datasets composed of either WOFS or FS acquisitions. Zhang et al. [13] investigated instead a transfer learning approach to segment FS acquisitions using a network trained on WOFS acquisitions. Fashandi et al. [14] looked at the effect of different inputs to the U-Net architecture for the task of breast segmentation only, concluding that a U-Net architecture trained with both FS and WOFS acquisitions stacked together on the channel dimension and passed as input to a U-Net architecture achieves the best performance.

In this work, we present a simple yet efficient means of exploiting the datasets available in order to make the segmentation network robust to different breast MRI T1 sequences, without the need of additional annotations or re-training. We follow the approach of Huo et al. [11], employing the nnUNet framework, and by using a dataset of breast DCE-MRI acquired via the Dixon method, we train a 3D U-Net on both FS and WOFS acquisitions with the same ground truth annotations. We found that this approach allows the network to robustly handle both FS and WOFS acquisitions at inference time, without explicit information about the type of image, while needing only half the amount of manual annotation for training. We validated our approach both quantitatively in our internal test set and by means of a reader study on both the internal test set and an external, publicly available dataset. Our algorithm is publicly available for personal testing and visualization of the results on users’ own breast MRI acquisitions (https://grand-challenge.org/algorithms/breast-and-fibroglandular-tissue-segmentation, accessed on 29 May 2022).

## 2. Materials and Methods

### 2.1. Internal Dataset

In this retrospective study, 40 patients were selected from the PACS system of our institution (Radbodumc, Nijmegen, The Netherlands) based on the availability of breast DCE-MRI acquisitions acquired with the use of the Dixon method. All the scans were acquired between November 2015 and February 2017. Patients (mean age 50 ± 21 years, all females) were all participating in an increased-risk breast cancer screening program that consisted of annual MRI with or without concurrent mammography, with no previous history of breast cancer. All breast MRI acquisitions were obtained using a 3T breast MRI scanner (Siemens Trio, Prisma or Skyra) and at least a 16-channel breast coil. Breast MRI acquisition protocols have slightly changed over time, but all met the minimal standards described within the EUSOBI breast MRI guideline. Table 1 shows an overview of the acquisition parameters. All the acquisitions included in this study were pre-contrast T1-weighted 3D Dixon-type acquisitions. Only the in-phase and the reconstructed water acquisitions were used from the four available (in-phase, out-of-phase, reconstructed water and fat) Dixon outputs. The in-phase and the reconstructed water acquisitions, respectively, are WOFS (Figure 1a) and FS (Figure 1b) acquisitions. We divided the dataset into training (n = 31 patients, 62 acquisitions (31 WOFS, 31 FS) and testing (n = 9 patients, 18 acquisitions (9 WOFS, 9 FS)). The testing set was held out completely from the implementation of the segmentation network and it was used only for the final validation.

#### Ground Truth Annotations

The breast and FGT regions were manually annotated by R.S. under the supervision of an experienced breast radiologist (with 16 years of experience) using the software ITKSNAP [15]. The breast fat tissue and FGT regions were manually annotated using the WOFS image—see Figure 2—and, when in doubt about boundaries for either breast fat tissue or FGT areas, also the FS image was consulted. The exact same segmentation could be used for WOFS and FS acquisitions without the need of registration.

### 2.2. External Dataset

For further validation of the robustness of our approach not only on FS and WOFS acquisitions but also to different MRI scanner and parameter acquisitions, we used the Breast DCE-MRI dataset released publicly by Duke University in collaboration with the Cancer Imaging Archive [16]. This dataset comprises 922 patients with biopsy-confirmed invasive cancer. The breast MRI acquisitions were acquired with either 1.5 T or 3 T scanners in prone positions. The acquisitions available for each patient are: a WOFS T1 acquisition, a FS pre-contrast T1 acquisition and three to four FS post-contrast T1 acquisitions. For our validation, we used only the WOFS T1 acquisition and the FS pre-contrast T1 acquisition. Of the 922 patients available, we randomly selected 30 patients for our reader study. The random selection process guaranteed an equal distribution of MRI scanner and parameter acquisitions so as to be able to analyze the impact of acquisition parameters on the final segmentation quality. Table 2 shows an overview of the acquisition parameters and the number of samples used.

### 2.3. Neural Network Architecture

The automatic segmentation model used in this study is based on the nnUNet framework. The nnUNet or “no new U-Net” framework was created with the idea of having a standardized backbone to perform medical imaging segmentation based on the well-established U-Net architecture, first introduced by Ronneberger et al. [6], and subsequently extended into 3D [17]. The framework has already been validated on a wide range of medical imaging domains thanks to the success achieved in the Medical Segmentation Decathlon challenge [18]. The framework follows some pre-defined heuristic strategies to adapt pre- and post-processing steps (e.g., voxel spacing, normalization, etc.) based on the dataset at hand, as well as the U-Net configuration (e.g., network layers, batch size, etc.) to fit the requirements of the computational resources available. In addition, the framework applies extensive data augmentation (e.g., rotation, cropping) at training time to achieve better generalization. In this study, we used the patch-based full-resolution 3D U-Net as the starting point for our experiments. We compared a segmentation network trained with only WOFS acquisitions with a network trained with both WOFS and FS acquisitions. For the training with both WOFS and FS acquisitions, we treated each image as a separate sample but we used the same annotation. The input of the network was always one patch of size 192 × 192 × 64 and a batch size of 2. Both networks were trained for the pre-defined number of 1000 epochs, which was chosen from the nnUNet framework as showing the best results.

### 2.4. Experiments and Evaluation

#### 2.4.1. Quantitative Evaluation with Ground Truth Annotations

In our first experiment, we compared a network trained with WOFS-only acquisitions and a network trained with WOFS + FS acquisitions. The held-out internal test set (n = 9 patients, 18 acquisitions (9 FS and 9 WOFS)) was used to quantitatively assess the quality of the segmentation produced by both networks. We used the Dice Similarity Coefficient (DSC) as measure to compare the manual annotation ground truth with the segmentation generated by the two different networks. Furthermore, we calculated 95% Confidence Intervals (CI) on DSC for each category using the bootstrapping method with 15,000 repetitions. Subsequently, we used Bland–Altman plots for the best-performing network to analyze the agreement between the breast density calculated using the ground truth manual annotation and the automatic segmentation.

#### 2.4.2. Reader Study for Visual Assessment

In our second experiment, we used the best-performing network from the previous experiment to validate the robustness of the segmentation on an external dataset with heterogeneous acquisition parameters and scanner types. In order to validate the results achieved in our internal test set, and due to the lack of externally publicly available datasets with ground truth annotations, we set up a reader study using the Grand Challenge platform [19].

We asked two breast radiologists (16 and 5 years of experience) to visually assess the quality of the breast and FGT segmentation produced by the network trained with WOFS + FS acquisitions. In Figure 3, an example of the reader study set-up is shown. For each case included in the reader study, we presented, side-by-side, both the FS and WOFS breast MRI acquisition with an overlay produced by the segmentation network for both acquisitions. The participant had the possibility to scroll through the volumes simultaneously and adjust the segmentation overlay transparency. The radiologist was asked to rate the quality of the breast and FGT segmentation both in the FS and WOFS versions of the acquisition for a total of four questions. The rating consisted of a four-point rating scale (1—Poor, 2—Fair, 3—Good, 4—Excellent). Additionally, the radiologist was asked to express a preference for the overall segmentation quality between the segmentation for the FS image and WOFS image, with the option of indicating no significant difference in quality. The order in which the cases were presented was randomized to account for a possible learning bias error, for every radiologist participating in the reader study. There was no restriction of time to finalize the reader study, and the radiologist had the possibility of partially completing the reader study before resuming from the point at which they left. Radiologists participating in the reader study had only access to the FS and WOFS acquisitions for each case, without any other clinical information.

To statistically evaluate the results of the reader study, we calculated average rating scores and 95% CIs for each reader using the bootstrapping method with 15,000 repetitions, sub-dividing the rating per category (breast or FGT) and per acquisition type (WOFS or FS). Furthermore, we assessed inter-reader agreement using percentage agreement. Other methods of inter-reader agreement (e.g., Cohen’s Kappa coefficient) could not be applied due to the nature of some of our results.

## 3. Results

### 3.1. Quantitative Evaluation with Ground Truth Annotations

The quantitative evaluations against the manual annotation for both the network trained on WOFS-only acquisitions and the network trained with WOFS + FS acquisitions are shown in Table 3.

The network trained on WOFS only achieved 0.96 and 0.92 DSC, respectively, for breast and FGT segmentation of WOFS test acquisitions. For FS test acquisitions, however, the same network failed to generalize, achieving 0.1 and 0.15 DSC on breast and FGT segmentation, respectively. The network trained with WOFS + FS acquisitions, instead, performed with a similar DSC score on both types of acquisitions at test time, achieving 0.96 and 0.91 DSC for breast and FGT on WOFS test acquisitions and 0.95 and 0.86 DSC on FS test acquisitions.

Furthermore, we analyzed the difference between manual annotation ground truth breast density and breast density calculated from the segmentation produced by the network trained with WOFS + FS acquisitions, using a Bland–Altman plot on the internal test set. On both types of acquisitions, the difference was close to 0 (Figure 4), with a mean of −0.33 for WOFS acquisitions (Figure 4a) and a mean of −0.2 for FS acquisitions (Figure 4b).

### 3.2. Reader Study for Visual Assessment

For the internal dataset, both breast and FGT segmentations were consistently scored as having excellent quality by both breast radiologists. Figure 5 confirms the quantitative evaluation from Table 3 for the network trained with WOFS + FS acquisitions. The analysis of the average rating for individual readers in Table 4 and the percentage agreement between readers in Table 5 further validate the quality of the segmentations for the internal test set, reporting both high average ratings and almost perfect agreement between readers. The visual evaluation results for the external dataset are reported in Figure 6. Overall, the segmentation quality for both breast and FGT was considered of high quality by both breast radiologists. The highest loss in quality could be noticed for the FGT segmentation in the FS acquisitions. As for the internal test set, we analyzed the average rating for individual readers with 95% CIs, reported in Table 6, and the percentage agreement between readers, reported in Table 7. From Table 6, we can see a difference in average ratings between the two readers, with one of the readers consistently rating the acquisitions of higher quality. The consistently higher scoring from one of the readers has a direct influence on the percentage agreement, which remains overall substantial between the two readers, with the strongest disagreement on the ratings for the breast segmentation in FS acquisitions. Figure 7 shows some examples of the FGT segmentations that were rated as either fair or poor by the radiologists.

## 4. Discussion

Using a simple data-centric approach, we were able to build a segmentation network robust to both breast MRI T1 WOFS and FS acquisitions, without the need of double annotations for both types of acquisitions. Our results are in line with what was partially found by Fashandi et al. [14] for the task of breast segmentation only. Compared to [14], our choice of a dataset with Dixon acquisitions allowed us to skip the registration of the WOFS and FS acquisitions and still use the same annotation, and we expanded their findings by showing robustness also on the more challenging task of FGT segmentation. Furthermore, we did not stack WOFS and FS acquisitions on two channels as input, to allow the network to be used in situations where either the FS or WOFS acquisitions are not available. This choice of single-channel input might have sacrificed some performance for better generalizability. Our reader study has further validated the robustness of the network on an external dataset with different acquisition parameters.

Compared to Huo et al. [11], we are using a one-stage approach with three classes: background, breast and FGT. The advantage of this approach compared to a two-stage approach is the faster inference and the fine-tuning of a single network if re-training is needed. Adding a penalty term to the loss function when FGT segmentation is predicted outside the breast contour might eliminate the need of a two-stage approach completely. We did not explore this option, but it might be an useful direction for further research.

We did not report quantitative comparisons with other approaches for breast and FGT segmentation because DSC might potentially be biased towards the way in which the annotation process was performed. The manual annotation of breast and FGT can have great inter- and intra-reader variability because there are no clearly defined boundaries to perform this task. In general, we believe that a reader study with a panel of expert breast radiologists is a better means of validating the quality of the segmentation for this particular problem. The conducted reader study aimed at assessing the general quality of the segmentation according to radiologists and gives a better look at the final clinical application of the approach. Because the segmentations were consistently rated of good quality, we presume that radiologists will also be more willing to use the automatic software in clinical practice and accept the results of this otherwise tedious task. The study presented here has several limitations. The first limitation is the relatively small size of the internal dataset used for training/validation and testing. The purpose of this study was not to achieve the highest possible performance in terms of DSC or other quantitative segmentation metrics, but it was to show the robustness of the approach on both FS and WOFS acquisitions while retaining reasonable performance. For this reason, and by following general guidelines found in the literature for the optimal size of the training samples [7,14], we decided to use a relatively small dataset. A second limitation of our approach is the way in which the annotations were created for the internal dataset. All manual annotations were performed by a single person and reviewed by one breast radiologist, but using an average of multiple expert annotations for the same volumes may result in a better estimate of the real ground truth annotation. A future direction of research might also be to investigate whether it is possible to use partial pixel annotation for the training of the segmentation network, which would allow the relatively fast extension of an annotated dataset, which might also be more reproducible. As the internal dataset was from a single center and mostly acquired with the same parameters from a similar type of scanner, a larger, multi-center, multi-vendor dataset, using the same strategy as presented in this manuscript, may yield an even more robust segmentation network. Finally, further external validation may be valuable. However, there are not a lot of publicly available breast MRI datasets with both WOFS and FS acquisitions.

## 5. Conclusions

In conclusion, we created a stable segmentation algorithm that can be used to estimate breast density from any of the most common types of T1-weighted breast MRI acquisitions. Based upon a reader study, the algorithm provided excellent results in an internal test set and also performed well in an external validation. Further extension of the training sets may further improve the quality of the segmentations.

## Figures and Tables

**Figure 1 diagnostics-12-01690-f001:**
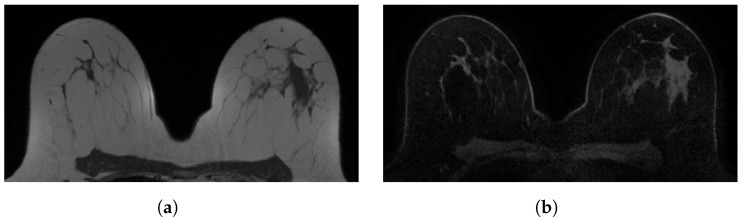
Examples of in-phase and reconstructed water Dixon acquisitions from our internal dataset used to mimic acquisitions without (WOFS) and with (FS) fat suppression. (**a**) In-phase Dixon acquisition used as WOFS image. (**b**) Reconstructed water Dixon acquisition used as FS image.

**Figure 2 diagnostics-12-01690-f002:**
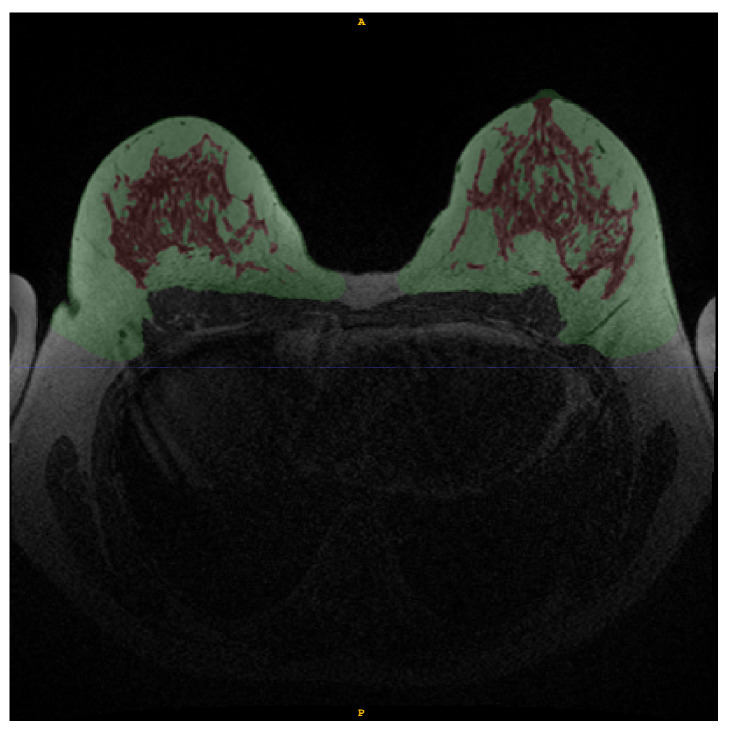
Example of manual annotation of fat tissue (green) and FGT (red) in WOFS image.

**Figure 3 diagnostics-12-01690-f003:**
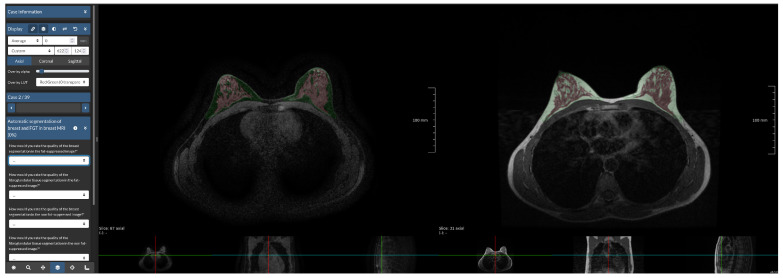
Screenshot of reader study set-up. The Grand Challenge platform allows the set-up of a web-based reader study with a fully functional medical imaging viewer. In this reader study, we showed participants the same WOFS and FS case side-by-side, and they were asked to answer the questions on the left side of the screen, which, in this case, were multiple choice questions.

**Figure 4 diagnostics-12-01690-f004:**
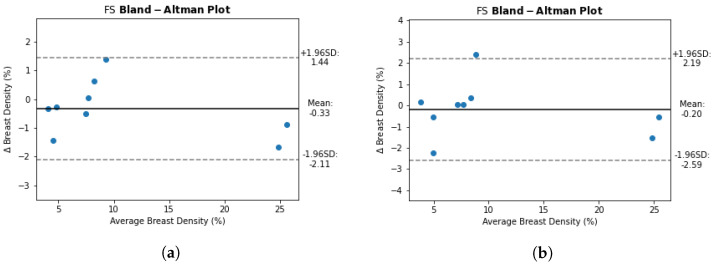
Bland–Altman plot to compare the breast density (BD) calculated on ground truth manual annotation and BD calculated from breast and FGT segmentations generated by the network trained with WOFS + FS acquisitions. (**a**) Bland–Altman plot for WOFS test cases. (**b**) Bland–Altman plot for FS test cases.

**Figure 5 diagnostics-12-01690-f005:**
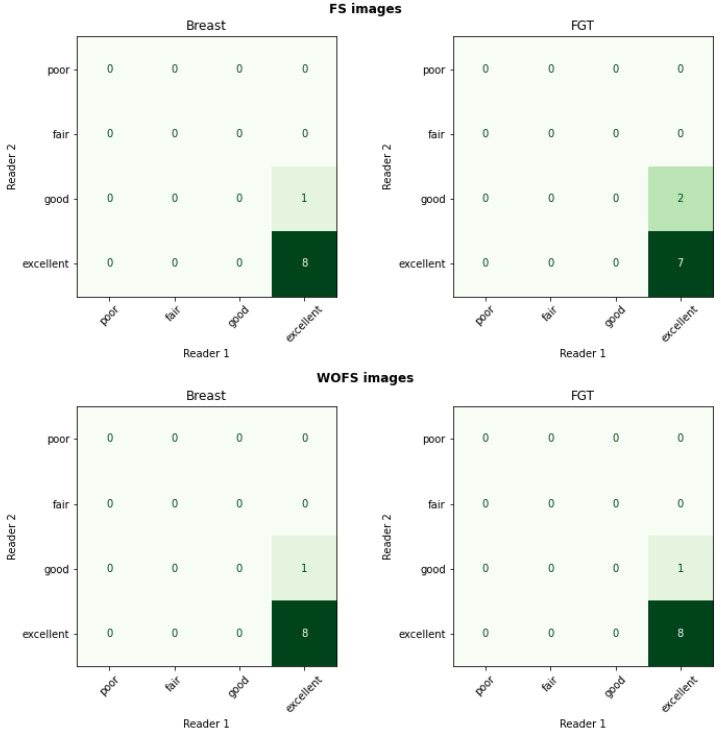
Internal test set results (n = 9 patients, 18 acquisitions (9 WOFS, 9 FS)). Reader study confusion matrix comparing the ratings (1—Poor, 2—Fair, 3—Good, 4—Excellent) from the two readers.

**Figure 6 diagnostics-12-01690-f006:**
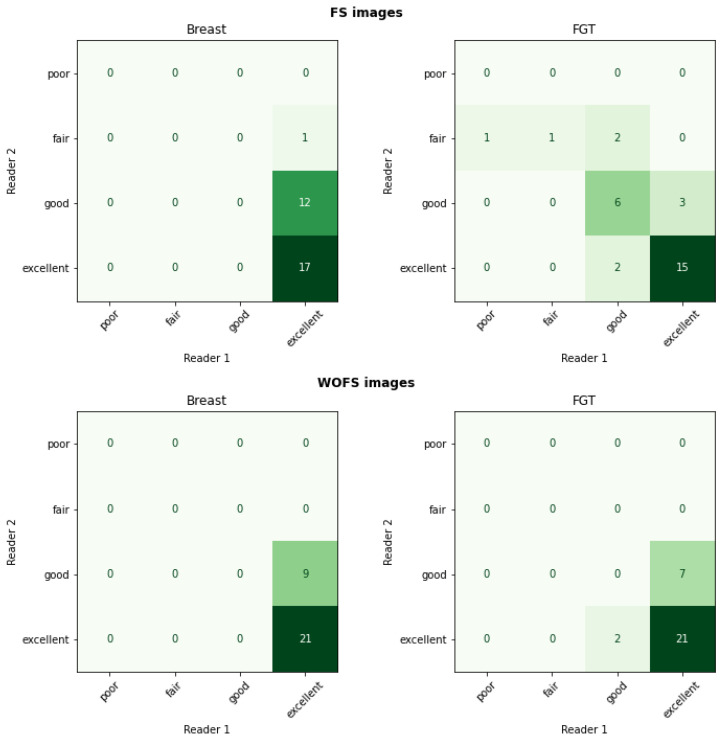
External test set results (n = 30 patients, 60 acquisitions (30 WOFS, 30 FS)). Reader study confusion matrix comparing the ratings (1—Poor, 2—Fair, 3—Good, 4—Excellent) from the two readers.

**Figure 7 diagnostics-12-01690-f007:**
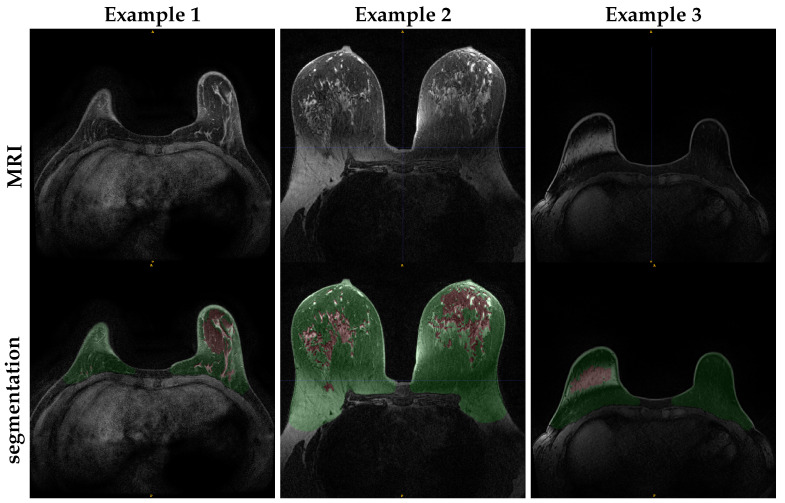
Example slices of FGT segmentations from external test set acquisitions rated either poor or fair during the reader study assessment.

**Table 1 diagnostics-12-01690-t001:** Breast DCE-MRI acquisition parameters for internal dataset.

Acquisition Parameter	
Repetition Time (ms)	4–5.68
Echo Time (ms)	2–2.46
Flip Angle	15–20
Magnetic Field Strength	3.0
Slice Thickness (mm)	1–2.5
Pixel Spacing (mm${}^{2}$)	(0.8 × 0.8)–(0.98 × 0.98)

**Table 2 diagnostics-12-01690-t002:** MRI scanner information from the randomly selected subset of the external validation dataset.

Manufacturer	Magnetic Field Strength (Tesla)	Model	Total (n = 30)
GE Medical Systems	1.5	Optima MR450w	5
		Signa HDx	3
		Signa HDxt	2
	3.0	Signa HDx	2
		Signa HDxt	3
Siemens	1.5	Avanto	5
	3.0	Skyra	5
		TrioTim	5

**Table 3 diagnostics-12-01690-t003:** Internal test set results (n = 9 patients, 18 acquisitions (9 WOFS, 9 FS)). Breast and FGT Dice Similarity Coefficients (DSC) and 95% Confidence Intervals (CI) from network trained only without fat suppression (WOFS only) acquisitions and without and with fat suppression acquisitions (WOFS + FS).

Network Trained on	Test	Breast DSC [95% CI]	FGT DSC [95% CI]
WOFS only	WOFS	0.96 [0.94, 0.97]	0.92 [0.89, 0.95]
	FS	0.10 [0.08, 0.12]	0.15 [0.11, 0.19]
WOFS + FS	WOFS	0.96 [0.95, 0.97]	0.91 [0.87, 0.94]
	FS	0.95 [0.94, 0.96]	0.86 [0.82, 0.91]

**Table 4 diagnostics-12-01690-t004:** Internal test set results (n = 9 patients, 18 acquisitions (9 WOFS, 9 FS)). Reader study average rating (1—Poor, 2—Fair, 3—Good, 4—Excellent) and 95% Confidence Intervals (CI) per reader.

		Breast		FGT	
	**Acquisitions Type**	**Avg. Rating**	**95% CI**	**Avg. Rating**	**95% CI**
**Reader 1**	WOFS	3.89	[3.67, 4.0]	3.89	[3.67, 4.0]
	FS	3.89	[3.67, 4.0]	3.78	[3.44, 4.0]
**Reader 2**	WOFS	4.0	[4.0, 4.0]	4.0	[4.0, 4.0]
	FS	4.0	[4.0, 4.0]	4.0	[4.0, 4.0]

**Table 5 diagnostics-12-01690-t005:** Internal test set results (n = 9 patients, 18 acquisitions (9 WOFS, 9 FS)). Reader study overall average rating (1—Poor, 2—Fair, 3—Good, 4—Excellent) and percentage agreement between readers.

	Breast		FGT	
**Acquisitions Type**	**Avg. Rating**	**% Agreement**	**Avg. Rating**	**% Agreement**
WOFS	3.94	89%	3.94	89%
FS	3.94	78%	3.88	89%

**Table 6 diagnostics-12-01690-t006:** External test set results (n = 30 patients, 60 acquisitions (30 WOFS, 30 FS)). Reader study average rating (1—Poor, 2—Fair, 3—Good, 4—Excellent) and 95% Confidence Intervals (CI) per reader.

		Breast		FGT	
	**Acquisitions Type**	**Avg. Rating**	**95% CI**	**Avg. Rating**	**95% CI**
**Reader 1**	WOFS	3.7	[3.53, 3.87]	3.77	[3.6, 3.9]
	FS	3.53	[3.33, 3.73]	3.43	[3.17, 3.67]
**Reader 2**	WOFS	4.0	[4.0, 4.0]	3.93	[3.83, 4.0]
	FS	4.0	[4.0, 4.0]	3.5	[3.23, 3.73]

**Table 7 diagnostics-12-01690-t007:** External test set results (n = 30 patients, 60 acquisitions (30 WOFS, 30 FS)). Reader study overall average rating (1—Poor, 2—Fair, 3—Good, 4—Excellent) and percentage agreement between readers.

	Breast		FGT	
**Acquisitions Type**	**Avg. Rating**	**% Agreement**	**Avg. Rating**	**% Agreement**
WOFS	3.85	70%	3.85	70%
FS	3.77	57%	3.47	73%

## Data Availability

The internal dataset used for this study can be made available for reproducibility or comparison purposes under a formal written exchange agreement with our institution.

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
