# Peer review of "Exploiting the Dixon Method for a Robust Breast and Fibro-Glandular Tissue Segmentation in Breast MRI"

_diagnostics, 2022, doi:10.3390/diagnostics12071690_

Round 1

Reviewer 1 Report

The authors did a great job explaining their methods and results. Some minor edits/suggestions.

1. Some grammatical corrections are necessary. For example, Tense needs to be revised on line 122.

2. What's the difference between two Signa HDx/Signa HDxt in table 2?

3. What's the meaning of X years of experience on line 147. I think it's not necessary to include the duration of the experience.

4. Need more details for figure legend (figures 6 and 7).

Also, I think it will be better to include more details about the patients (i.e., demographics, treatment).

Best of luck.

Author Response

Point 1: Some grammatical corrections are necessary. For example, Tense needs to be revised on line 122.

Response 1: Thank you for spotting the mistake. The revised manuscript has been proof-read again and, hopefully, all the mistakes have been corrected.

Point 2: What's the difference between two Signa HDx/Signa HDxt in table 2?

Response 2: The Signa HDxt is an upgraded version of the Signa HDx. Honestly, I am not sure if, from an acquisition point of view, there is any difference. We read all the information directly from the dicom metadata and we reported them, as is, for completeness.

Point 3: What's the meaning of X years of experience on line 147. I think it's not necessary to include the duration of the experience.

Response 3: In the literature, the years of experience are normally reported when either a reader study or a segmentation labeling task is conducted. From my understanding, the years of experience are reported to substantiate the trustworthiness of the task that has been performed (in our case, the correctness of the segmentation for the breast and the fibroglandular tissue, given the fact that it is a specialized task). X years of experience in our manuscript is a typo mistake where we forgot to insert the real years of experience of the breast radiologist. The revised version of the manuscript includes the correct years of experience for both radiologist who took part in our reader study.

Point 4: Need more details for figure legend (figures 6 and 7)

Response 4: The description for both figures have been updated in the revised manuscript and hopefully better clarifies the figure.

Point 5: Also, I think it will be better to include more details about the patients (i.e., demographics, treatment).

Response 5: We welcomed the feedback to improve our manuscript and we included some additional information in the Methods and Materials section about the population of our internal dataset. For the external dataset, we cited the corresponding journal paper.

We would like to kindly thank the reviewer for taking the time to read our manuscript and give suggestions on how to improve it. Hopefully, we addressed all the concerns the reviewer had about our manuscript but we remain available for any further question or concern.

Reviewer 2 Report

The abstract needs quantification. Section 2 needs more clarification. The discussion part needs improvement. Technical Soundness of segmentation can be verified through mathematical models. The conclusion needs modification.

The objectives are clear but the presentation needs improvement.

Author Response

Point 1: The abstract needs quantification.

Response 1: In the abstract of our revised manuscript we added key findings and results which should address the request for quantification.

Point 2: Section 2 needs more clarification. The discussion part needs improvement. Technical Soundness of segmentation can be verified through mathematical models. The conclusion needs modification.

Response 2: We grouped the rest of the comments of the reviewer in a single point because we think that most of the concerns have been addressed in response to several points raised by the other reviewer and the editor. The revised manuscript includes a description of the statistical analysis that has been added to validate the results of the quantitative evaluation and the reader study. Description of tables and figures have been improved. Hopefully all these modification address also the points raised here.

We would like to kindly thank the reviewer for taking the time to read our manuscript and give suggestions on how to improve it. Hopefully, we addressed all the concerns the reviewer had about our manuscript but we remain available for any further question or concern.